synthetic chemistry

polyimides, microwave, pyridine, transparency, thermal properties

**Authors for correspondence:**
Yue-qi Mo
e-mail: pomoy@scut.edu.cn
Jian-qing Zhao
e-mail: psjqzhao@scut.edu.cn

This article has been edited by the Royal Society of Chemistry, including the commissioning, peer review process and editorial aspects up to the point of acceptance.

# Microwave-assisted synthesis of high thermal stability and colourless polyimides containing pyridine

Kai Cheng, Jie-pin Hu, Yan-cheng Wu, Chu-qi Shi, Zhi-geng Chen, Shu-mei Liu, Yan-chao Yuan, Yue-qi Mo and Jian-qing Zhao

School of Materials Science and Engineering, South China University of Technology, Guangzhou, Guangdong 510640, People's Republic of China

(iD) Y-qm, 0000-0002-6322-7072

A novel aromatic diamine containing pyridyl side group, 4-pyridine-4,4-bis(3,5-dimethyl-5-aminophenyl)methane (PyDPM), was successfully synthesized via electrophilic substitution reaction. The polyimides (PIs) containing pyridine were obtained via the microwave-assisted one-step polycondensation of the PyDPM with pyromellitic dianhydride (PMDA), 3,3′,4,4′-biphenyltetracarboxylic dianhydride (BPDA), 3,3′,4,4′-diphenylether tetracarboxylic dianhydride (ODPA) and 4,4′-(hexafluoroisopropylidene)diphthalic anhydride (6FDA). Contrarily to the reported similar PIs, these PIs exhibit much higher thermal stability or heat resistance, i.e. high glass transition temperatures ($T_g$s) in the range of 358–473°C, and the decomposition temperatures at 5% weight loss over 476°C under nitrogen. They can afford flexible and strong films with tensile strength of 82.1–93.3 MPa, elongation at break of 3.7%–15.2%, and Young's modulus of 3.3–3.8 GPa. Furthermore, The PI films exhibit good optical transparency with the cut-off wavelength at 313–366 nm and transmittance higher than 73% at 450 nm. The excellent thermal and optical transmittance can be attributed to synthesis method and the introduction of pyridine rings and ortho-methyl groups. The inherent viscosities of PIs via one-step method were found to be 0.58–1.12 dl g$^{-1}$ in DMAc, much higher than those via two-step method. These results indicate these PIs could be potential candidates for optical substrates of organic light emitting diodes (OLEDs).

## 1. Introduction

With the rapid development of the optoelectronic field and microelectronics manufacturing industry, substrates with high

**Scheme 1.** Some reported TPM-derivatives.

glass transition temperature and high transmittance have attracted significant attention. The ideal optically colourless polymer film requires stability at high processing temperatures, high transparency in the visible range, and good mechanical properties. For example, in the manufacture of organic light emitting devices (OLEDs), the processing temperature on flexible polymer films might exceed 300°C [1]. Most of common transparent polymers such as PS, PC, PET, PEN with glass transition temperature, $T_g < 150$°C obviously cannot tolerate such high temperature. Aromatic polyimides are high-performance polymers with excellent thermal and mechanical properties and have been widely applied in microelectronics, aerospace and separation techniques etc. [2,3]. However, conventional aromatic polyimide (PI) films exhibit dark colours and poor transparency in the visible region arising from the charge–transfer (CT) interactions, and hence they cannot be used as the plastic substrates in OLED displays and so on. Furthermore, the conventional aromatic PIs are always insoluble in common solvents due to their rigid chain characteristics and the strong molecular interactions, which limits their applications in many areas. Thus, the colourless PIs with high $T_g$ and high solubility [4] are highly valuable at present.

In order to obtain a light or completely colourless PI, it is necessary to reduce the charge transfer complexes (CTC) formation among PI chains. In the past 30 years, many efforts have been devoted to modifying PIs at the molecular level by the introduction of flexible linkages [5], asymmetric units [6], bulky side groups [7], noncoplanar structures [8], alicyclic moieties [9] and fluorinated substituents [10]. Fluorination is one of the most effective methods to simultaneously improve soluble, optical, hydrophobic and dielectric performances of PIs because of the strong electronegativity and low molar polarizability of fluorine atom [11]. However, the low-surface energy of the fluorine atoms brings low adhesion properties, which limits their applications. Another efficient approach to colourless PIs is to introduce flexible spacer units such as ether [12] or substituted methylene [13]. Among them, 4,4′-diaminotriphenylmethane (TPM) and its derivatives have been employed as superior monomers to diaminodiphenylmethane (DDM) because they have one more bulky phenyl ring. Likhatchev *et al.* [14] first reported TPM-based PIs and found that these soluble PIs showed relatively high $T_g$s, e.g. PI composed of TPM and pyromellitic dianhydride (PMDA) (TPM-PMDA) shows $T_g$ of 365°C. Subsequently, people explored a series of TPM derivatives (as shown in scheme 1) by introducing various aromatic units into $R_1$ position and aliphatic units into the $R_2$ and $R_3$ positions of the diamine

[15–24]. $R_1$ groups were found to have great effect on the $T_g$s of the final polymers. Hamciuc *et al.* [25] prepared several novel TPM derivatives, such as phthalonitrile diamine (PA) and 4,4′-(naphthalen-1-ylmethylene)dianiline (NM), as shown in scheme 1. The PIs containing PA showed excellent solubility but low $T_g$s of 241–269°C while PIs containing NM showed much higher $T_g$s. For example, NM-BPDA showed $T_g$ of 341°C, much higher than that of TPM-BPDA (315°C) [26,27]. Moreover, most of the PIs containing TPM are still soluble in common solvents. On the other hand, $R_2$ and $R_3$ groups also play an important role in the thermal properties of PIs. Cheng *et al.* [28] found that methyl groups at the ortho-positions of the aniline ring can increase the $T_g$ of PIs; maybe it is because the introduced methyl groups can hinder the rotation of the two aromatic rings around the C–N bond. Li *et al.* [26] also found that PI composed of 4,4′-(naphthalen-1-ylmethylene) bis(2,6-dimethylaniline) (NMM) and BPDA (NMM-BPDA) has $T_g$ of 387°C, much higher than that of NM-BPDA (341°C). However, the $T_g$s of TPM-based PIs were not high enough compared with the commercial PI (Kapton®, $T_g = 390$°C). Furthermore, the high cost and challenge of the synthesis of TPM derivatives also obstruct their commercialization at present.

Pyridine is a basic heterocyclic compound structurally related to benzene. It is a polar and basic molecule due to the electronegative nitrogen atom. Pyridine is generally introduced into the PI main chain to increase the thermal decomposition temperature, glass transition temperature and chemical resistance [29]. The presence of nitrogen atom in the polymer structure offers a polarized bond which increases dipole–dipole interactions in the polymer–solvent system and improves the solubility of the resulting polymers [30]. Pyridine-containing polymers also show both high gas permeabilities and high selectivities towards $CO_2$, mainly due to Lewis acid–Lewis base electrostatic interactions between $CO_2$ and pyridinyl moiety [31,32]. However, recent studies do not support that the introduction of pyridine as side group can enhance the thermal properties of polymer [33–35]. For example, Koohmareh & Mohammadifard [34] prepared a series of PIs bearing bipyridine moieties, and these PIs not only showed an obvious decomposition at around 300°C, but also exhibited low $T_g$s ranging from 185°C to 240°C. Amutha & Sarojadevi [35] have synthesized several PIs with side pyridine groups (m-PyDPM, as shown in scheme 1) and obtained polymers exhibited relatively low $T_g$s of less than 317°C. Therefore, it is still a question whether the introduction of pendent pyridine groups would sacrifice the thermal properties of polymers. This encourages us to further investigate the effect of side pyridine groups.

In this article, PyDPM, a novel pyridine containing TPM derivative bearing dimethyl groups at $R_2$ and $R_3$ positions, was prepared considering that the introduction of ortho-methyl substituents can effectively enhance the $T_g$s of the final polymers. All the polymers were synthesized by one-step polymerization of PyDPM with four commercial dianhydrides (PMDA, BPDA, ODPA and 6FDA) under microwave irradiation, because the traditional two-step method failed to get high-molecular-weight polymers. Contrarily to the literature, these PIs exhibit good thermal stability and high heat resistance, with $T_{d5}$ exceeding 467°C in nitrogen and $T_g$s ranging from 358°C to 473°C. Furthermore, all of the PIs including PyDPM-PMDA exhibit good solubility in polar and/or non-polar solvents. The cut-off wavelength of PI films except PyDPM-PMDA is in the range of 313–366 nm, and the transmittance at 450 nm is higher than 82%, indicating that these almost colourless PI films except PyDPM-PMDA can be potential candidates for optical substrates of OLEDs.

# 2. Experimental

## 2.1. Materials

PMDA (98%), ODPA (98%), BPDA (98%) and 6FDA (98%) were purchased from TCI and dried in vacuum at 120°C overnight before use. 2,6-dimethylaniline (99%), pyridine-4-aldehyde (99%), 2-methoxyethanol (99%) and isoquinoline (98%) were purchased from Aldrich and used as received. Concentrated hydrochloric acid and sodium hydroxide (AR) were purchased from Aladdin. *N,N*-dimethylformamide (DMF), *N,N*-dimethylacetamide (DMAc) and m-cresol were purified by vacuum distillation over calcium hydride prior to use. Other commercial solvents were analytical grade from Guangzhou Chemical Reagent Factory and used without further purification.

## 2.2. Characterization

[1]H NMR test was carried out on a Bruker Avance 400 spectrometer at resonant frequencies of 400 or 600 MHz, using DMSO-$d_6$ and CDCl$_3$ as solvent. Elemental analysis was tested on a CHNS element

analyser. Fourier-transform infrared spectra (FT-IR) and attenuated total reflection Fourier-transform infrared spectra (ATR FT-IR) were recorded with a Bruker Vector 22 spectrometer between 400 and 4000 cm$^{-1}$. High resolution liquid chromatography–mass spectrometry (HRLC-MS) data were recorded on an Agilent 1290-micrOTOF-QII instrument. Thermal stability was assessed using thermogravimetric analyser (TGA) operated at a 20°C min$^{-1}$ from 30 to 800°C under $N_2$ gas flow. Differential scanning calorimetry (DSC) analysis was conducted on a Netzsch DSC 200F3 at a heating rate of 5°C min$^{-1}$ under nitrogen atmosphere from 30 to 400°C and $T_g$ was obtained in the middle of the thermal transition from the second heating. The melting point of synthesized monomer was also measured using the DSC instrument at a heating rate of 5°C min$^{-1}$. Dynamic mechanical thermal analysis (DMA) was conducted on a TA Instruments DMA Q800 at a heating rate of 5°C min$^{-1}$ from 30 to 500°C and a load frequency of 1 Hz under nitrogen, and the specimen was made with 6 mm width, 25 mm length, 50 μm thickness. $T_g$ was determined as the peak temperature of tan $\delta$ curve. The coefficient of thermal expansion (CTE) was evaluated by using a Netzsch 402F3 thermal-mechanical analyser (TMA) from 30 to 200°C at a heating rate of 5°C min$^{-1}$ under nitrogen. Mechanical properties were evaluated using an Instron 5967 model. The strain rate was 10 mm min$^{-1}$ with at least five PI film specimens of $4 \times 25$ mm. The inherent viscosities ($\eta_{inh}$) of the solutions were measured with an Ubbelohde viscometer at $30 \pm 0.1$°C with a concentration of 0.5 g dl$^{-1}$ in DMAc. The number-average molecular weight ($M_n$) and polydispersity index (PDI) were obtained by gel permeation chromatography (GPC) relative to polystyrene standard on a US Waters 1515 instrument with DMF as eluent at a flow rate of 1.0 ml min$^{-1}$. Ultraviolet–visible (UV–Vis) spectra were recorded on a 3600 UV–Vis spectrophotometer at room temperature. Dielectric constants were investigated by a Hewlett-Packard 4284A dielectric spectrometer in the frequency range of 100 Hz to 10 MHz at room temperature. Wide-angle X-ray diffraction (WXRD) pattern was taken from 10° to 40° ($2\theta$ values) with Cu K$\alpha$ radiation. Water uptake of polyimide films was measured by the weight changes before and after immersion of the polyimide films in deionized water at 25°C for 24 h.

## 2.3. Monomer synthesis

### 2.3.1. Typical procedure of synthesis of PyDPM

2,6-dimethylaniline (48.47 g, 0.4 mol) was charged into a flask equipped with a reflux condenser and heated to 100°C under argon. Then a solution of pyridine-4-aldehyde (10.71 g, 0.1 mol), 2-methoxyethanol (25 ml) and concentrated hydrochloric acid (37 wt%, 16.7 ml) was added dropwise over an hour. The mixture was stirred at 125°C for another 12 h and cooled down. Then, an aqueous solution of sodium hydroxide (60 ml, 20 wt%) and methanol (500 ml) was added to afford a white solid. The crude product was filtered and washed thoroughly with deionized water, recrystallized from ethanol to produce white needles. Yield 22.20 g, 67.6%, mp 160°C. FT-IR (KBr): 3429, 3323 (N–H), 2960, 2860 (C–H), 1628, 1513, 1412, 1286 cm$^{-1}$. $^1$H NMR (DMSO-$d_6$) δ 8.37 (d, $J = 6.0$ Hz, 1H), 7.01 (d, $J = 6.0$ Hz, 1H), 6.49 (s, 4H), 5.07 (s, 1H), 4.39 (s, 4H), 1.97 (s, 12H). HRLC-MS (ESI): 332.21 (M + H)$^+$, Calcd 331.20 for $C_{22}H_{25}N_3$. Elemental analysis: calcd. for $C_{22}H_{25}N_3$: C, 79.72; H, 7.60; N, 12.68%. Found: C, 79.72; H, 7.79; N, 12.80%.

## 2.4. Polymer synthesis

### 2.4.1. Two-step method

PyDPM (0.6629 g, 2 mmol) was dissolved in 10 ml of dried DMAc. After the diamine was completely dissolved, 0.4362 g (2 mmol) of pyromellitic dianhydride was added in one portion. The reaction mixture was stirred at room temperature under nitrogen for 12 h to afford an intermediate polyamidic acid (PAA). PI films were prepared by casting the PAA solution onto glass plates with a heating programme (80°C/1 h, 120°C/1 h, 150°C/1 h, 180°C/0.5 h, 210°C/1 h, 240°C/0.5 h, 270°C/0.5 h, 300°C/0.5 h).

### 2.4.2. One-step method

A series of PIs containing pyridine were obtained via the microwave-assisted high-temperature polycondensation of PyDPM and four commercial dianhydrides, i.e. PMDA, BPDA, ODPA and 6FDA, respectively. In a typical experiment, the synthesis of polyimide PyDPM-PMDA, is given as

**Scheme 2.** Synthesis of diamine monomer PyDPM.

follows: a unique tube used for microwave reaction was charged with two monomers PyDPM (0.6629 g, 2 mmol), PMDA (0.4362 g, 2 mmol), 5 ml of m-cresol and about five drops of isoquinoline. All operations were carried out in a glove box under nitrogen atmosphere. The mixture was irradiated at 80°C for 2 h, 230°C for 30 min and a transparent solution was obtained. After cooling to room temperature, the viscous solution was diluted with appropriate $CHCl_3$ and precipitated by pouring into 500 ml ethanol and then filtered, washed with ethanol several times, and dried overnight under vacuum at 100°C to get the polymer PyDPM-PMDA. The PyDPM-BPDA, PyDPM-ODPA and PyDPM-6FDA were also obtained by a method similar to that discussed previously.

PyDPM-PMDA. Yield: 95%. IR (solution casting film, cm$^{-1}$): $\nu = 2963$ (CH$_2$-H), 1779 (C=O), 1731 (C=O), 1602, 1497, 1373 (C−N), 1185, 1109, 853, 809, 733. $^1$H NMR (400 MHz, CDCl$_3$) δ 8.53 (d, $J = 5.2$ Hz, 2H), 8.47 (s, 2H), 7.08 (d, $J = 5.2$ Hz, 2H), 6.89 (s, 4H), 5.41 (s, 1H), 2.03 (s, 12H).

PyDPM-BPDA. Yield: 97%. IR (solution casting film, cm$^{-1}$): $\nu = 2961$ (CH$_2$-H), 1777 (C=O), 1713 (C=O), 1593, 1484, 1362 (C−N), 1101, 853, 809, 733. $^1$H NMR (400 MHz, DMSO-$d_6$) δ 8.58 (d, $J = 5.2$ Hz, 2H), 8.48 (s, 2H), 8.42 (d, $J = 8.0$ Hz, 2H), 8.14 (d, $J = 8.0$ Hz, 2H), 7.29 (d, $J = 5.2$ Hz, 2H), 7.15 (s, 4H), 5.70 (s, 1H), 2.10 (s, 12H).

PyDPM-ODPA. Yield: 93%. IR (solution casting film, cm$^{-1}$): $\nu = 2923$ (CH$_2$-H), 1777 (C=O), 1714 (C=O), 1596, 1472, 1362 (C−N), 1103, 841, 816, 748. $^1$H NMR (600 MHz, DMSO-$d_6$) δ 8.54 (d, $J = 5.3$ Hz, 2H), 8.08 (d, $J = 8.0$ Hz, 2H), 7.72 (s, 2H), 7.67 (d, $J = 8.0$ Hz, 2H), 7.26 (d, $J = 5.3$ Hz, 2H), 7.11 (s, 4H), 5.66 (s, 1H), 2.06 (s, 12H).

PyDPM-6FDA. Yield: 93%. IR (solution casting film, cm$^{-1}$): $\nu = 2925$ (CH$_2$-H), 1721 (C=O), 1713 (C=O), 1595, 1485, 1365 (C−N), 1190, 1106, 720. $^1$H NMR (600 MHz, DMSO-$d_6$) δ 8.54 (s, 2H), 8.16 (s, 2H), 7.92 (m, 4H), 7.26 (s, 2H), 7.11 (s, 4H), 5.66 (s, 1H), 2.07 (s, 12H).

## 2.5. Membrane preparation

The PI resin was dissolved in DMAc with first stirring to give a homogeneous polyimide solution with a solid content of 10 wt% and then filtered through a 0.45 mm PTFE filter. The obtained homogeneous solution was poured onto a dry and clean glass plate. After that, the wet film was thermally baked in an oven at 80°C for 12 h and 200°C for 12 h to remove the solvent. The free-standing PI membrane with the thickness of 40–60 µm was prepared by immersing the glass plate in water followed by drying in an oven at 100°C for 12 h.

# 3. Results and discussion

## 3.1. Monomer synthesis and characterization

Firstly, we tried the reaction of pyridine-4-aldehyde with aniline and found its yield was quite low (approx. 30%) and failed to get the pure enough monomer. So we used 2,6-dimethylaniline instead of aniline as starting material. The aromatic diamines with pendent pyridyl group was successfully synthesized by electrophilic reaction between pyridine-4-aldehyde and 2,6-dimethylaniline catalysed by HCl, as shown in scheme 2. After recrystallization over ethanol, we can get pure enough PyDPM in a yield of 67.6%.

Figure 1 shows the $^1$H NMR spectrum of the compound PyDPM. The chemical shift of H$_1$ is observed at 8.37 ppm, much larger than its phenyl analogue TPMM [16], as shown in scheme 1. The protons in amino groups resonate at 4.39 ppm. Aromatic protons H$_4$ appear at 6.49 ppm as a single peak. The characteristic signals of methylidyne group, H$_3$, also shift to the low field at 5.07 ppm because of the

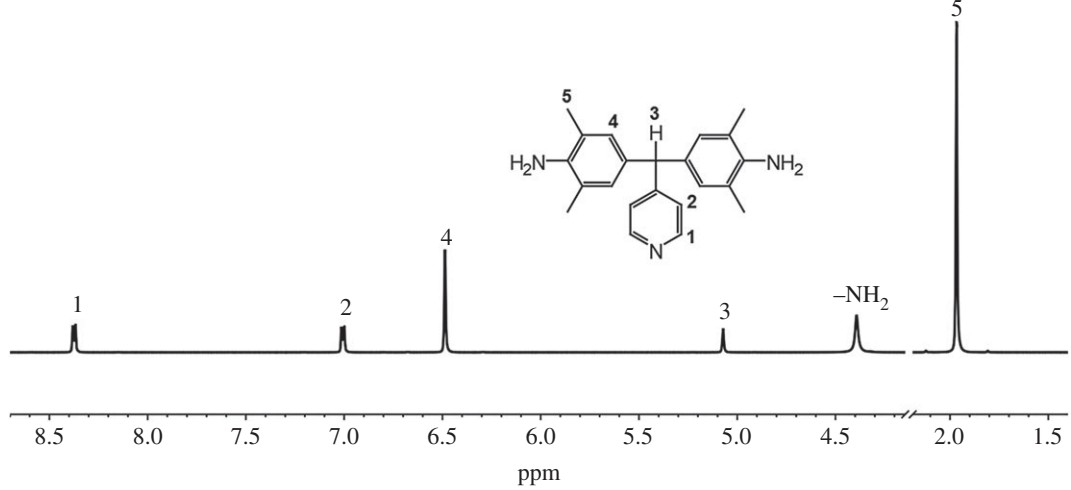

**Figure 1.** $^1$H NMR spectrum of PyDPM in DMSO-$d_6$.

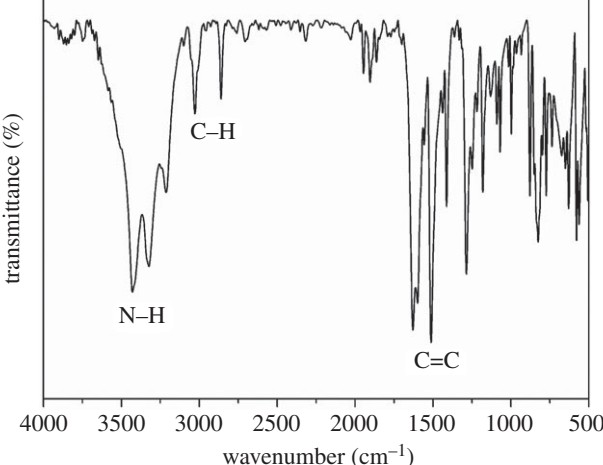

**Figure 2.** FTIR spectrum of PyDPM.

conjugative effect from the phenyl group. The signals assigned to the protons in the methyl groups, $H_5$, are indicated at 1.97 ppm.

The structure of PyDPM was further characterized by FTIR spectroscopy. The FTIR spectrum shows the characteristic absorption bands of the amino groups at 3323–3429 cm$^{-1}$ and the alkyl groups at 2860–2960 cm$^{-1}$. The characteristic C–H of triarylmethane appears at 3028 cm$^{-1}$. In the meantime, a strong absorption peak appears at 1628 cm$^{-1}$, which is caused by the vibration of the aromatic cycle and the deformation vibration of C=N. Moreover, the MS data and elemental analysis values are in good agreement with the proposed structures, demonstrating that the aromatic diamine PyDPM was successfully synthesized and the obtained monomer was pure enough to be employed for preparation of polyimides (figure 2).

## 3.2. Polymer synthesis and characterization

PI is usually prepared in two steps: first, polyamic acid is formed by ring-opening reaction at room temperature, and then is thermally cyclodehydrated at higher temperature. Here, the conventional two-step method could not obtain polymers with high molecular weight, which may be due to the great steric hindrance of methyl in the amino ortho-position. Thus, one-step high-temperature polycondensation procedure was used because it provides enough energy to ensure the smooth polymerization of monomers with lower activity and/or large steric hindrance substituents. Recently, microwave heating is being highly recognized by the advantages of higher temperature, shorter time, less side reactions and higher yield compared with conventional heating. Microwave-assisted

**Scheme 3.** Synthesis of the PIs.

**Table 1.** Inherent viscosities and GPC data of PIs.

| PIs | $\eta_{inh}$ (dl g$^{-1}$) | $M_n$ ($\times 10^4$ g mol$^{-1}$) | $M_w$ ($\times 10^4$ g mol$^{-1}$) | PDI |
|---|---|---|---|---|
| PyDPM-PMDA[a] | 0.20 | 0.60 | 1.10 | 1.83 |
| PyDPM-PMDA | 0.58 | 3.97 | 7.62 | 1.92 |
| PyDPM-BPDA | 1.12 | 6.13 | 12.18 | 1.98 |
| PyDPM-ODPA | 0.98 | 3.75 | 7.46 | 1.98 |
| PyDPM-6FDA | 0.59 | 6.37 | 12.55 | 1.97 |

[a]Two-step method.

synthesis of PIs also begin to attract great attention [26,36,37]. In this study, we made use of microwaves to synthesize PIs containing pyridine via one-pot polycondensation. Four commercial dianhydrides (PMDA, BPDA, ODPA and 6FDA) and diamine PyDPM were mixed with m-cresol and catalytic amount of isoquinoline in a glove box. Then, the mixture was transferred to a Biotage Initiator$^+$ synthesizer (400 W) and irradiated at 80°C for 2 h, subsequently 230°C for 30 min, which was automatically adjusted by the machine. After cooling to room temperature, the viscous solution was precipitated to afford four polymers PyDPM-PMDA, PyDPM-BPDA, PyDPM-ODPA, PyDPM-6FDA. The synthetic route is shown in scheme 3 and all the PI yields exceed 90%.

The data of inherent viscosities and GPC are listed in table 1. It was found that two-step method cannot produce PIs with high enough molecular weights. In fact, the cast films from PIs via two-step method readily break into pieces so that we cannot get more information from them. The PIs via one-step method show the inherent viscosities of PIs in the DMAc solution at 30°C are in the range of 0.58–1.12 dl g$^{-1}$, indicating that polymers have high molecular weights. The molecular weights of samples are further characterized by GPC. Their number-average molecular weights ($M_n$) and weight-average molecular weights ($M_w$) are recorded in the range of $3.97 \times 10^4$–$6.37 \times 10^4$ and $7.46 \times 10^4$–$12.55 \times 10^4$, respectively, and the PDI ($M_w/M_n$) is lower than 2.0. Obviously, these PIs present relatively high molecular weights and narrow molecular weight distributions. This may be related to microwave-assisted polymerization, which is favourable in increasing the molecular weight of polymers [26].

Figure 3 outlines the ATR FT-IR spectra of all the PIs. All the PIs exhibit characteristic absorption at 1780 cm$^{-1}$ (C=O asymmetrical stretching), 1720 cm$^{-1}$ (C=O symmetrical stretching) and 1367 cm$^{-1}$ (C–N stretching). The peaks at 2860–2960 cm$^{-1}$ are assigned to vibration of saturated alkyl group. The characteristic peak of triarylmethane appears at 3020 cm$^{-1}$, and the characteristic peak of the amino group at 3300–3500 cm$^{-1}$ disappears, indicating that all the pyridine-based PIs were almost imidized.

The representative $^1$H NMR spectrum of the PyDPM-BPDA is shown in figure 4. The peak at 7.2–8.7 ppm can be assigned to aromatic protons, and the characteristic proton of triarylmethane shifted from 5.07 to 5.70 ppm compared with its monomer PyDPM. The assignments of each proton designated in the $^1$H NMR spectrum are in complete agreement with the proposed polymer

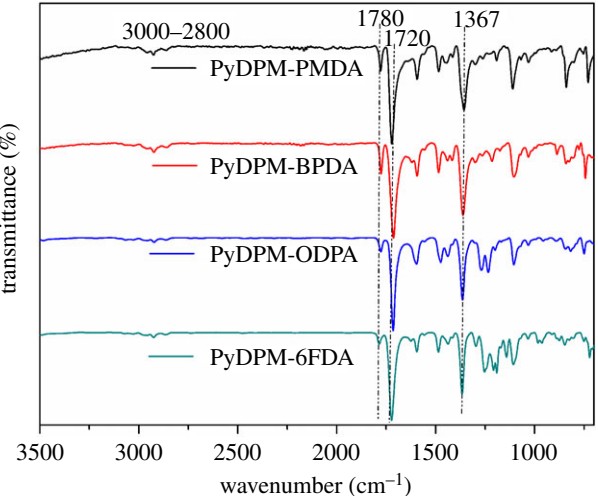

**Figure 3.** ATR FT-IR spectra of PI films.

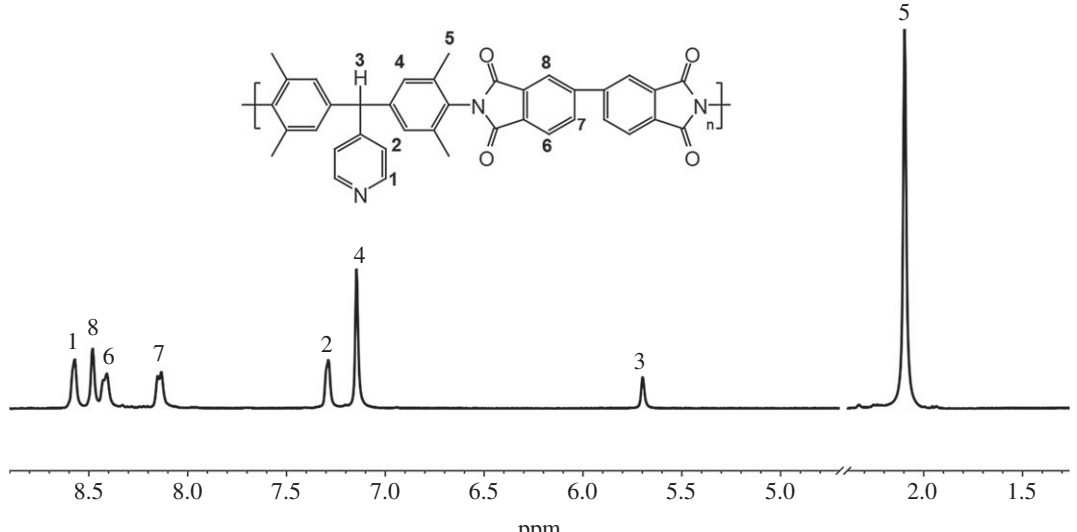

**Figure 4.** $^1$H NMR spectrum of PyDPM-BPDA in DMSO-$d_6$.

structures. These results indicate that the target PIs were successfully prepared by microwave-assisted one-step polycondensation, and the imidization was almost completed.

## 3.3. Polymer solubility

The solubility was measured by dissolving the PI films in different organic solvents with a test concentration of $10\,\text{mg ml}^{-1}$ and the results are summarized in table 2. It is well known that conventional aromatic PIs have poor solubility in ordinary solvents due to the strong interaction of molecular chains. For example, PIs polymerized from 4,4'-diaminodiphenylmethane (DDM) and anhydride 3,3',4,4'-benzophenonetetracarboxylic dianhydride (BTDA), ODPA are insoluble in ordinary solvents [38]. However, all of these PIs from PyDPM are readily soluble in polar solvents such as DMF, DMAc and m-cresol. Some of them are even soluble in less polar solvents with lower boiling points, such as $CHCl_3$ and THF. The excellent solubility of the obtained PIs was mainly attributed to the introduction of methyl groups and bulky pyridine rings, which formed noncoplanar configuration and decreased the packing density in the macromolecular chains. In acetone solvent testing, only PyDPM-6FDA can be dissolved rapidly at room temperature. This may be due to the incorporation of bulky hexafluoroisopropyl in the main chain, which further hindered the dense chain stacking. Contrarily to PyDPM-6FDA, PyDPM-PMDA exhibits inferior solubility and could not completely dissolve in DMSO,

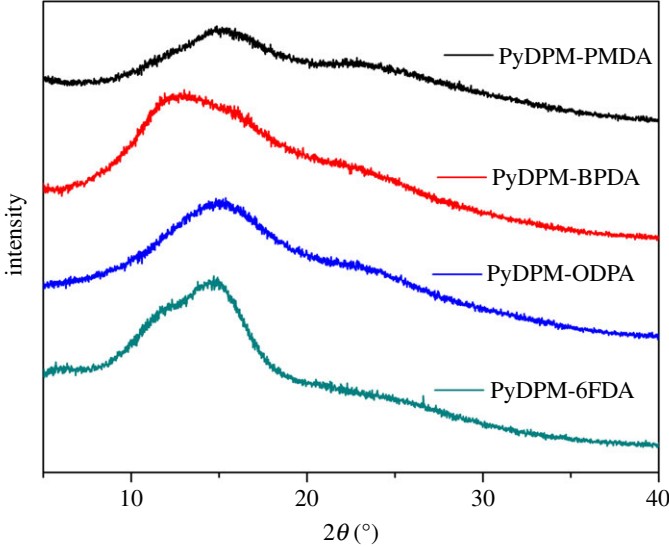

**Figure 5.** Wide-angle XRD patterns of PI films.

**Table 2.** Solubility of PIs. NMP: *N*-methyl-2-pyrrolidone; DMAc: *N,N*-dimethylacetamide; DMF: *N,N*-dimethylformamide; DMSO: dimethyl sulfoxide; THF: tetrahydrofuran; CHCl₃: trichloromethane. ++: soluble at room temperature. +: partially soluble at 60°C or the boiling point. −: insoluble.

| PIs | DMF | DMAc | DMSO | NMP | m-cresol | CHCl₃ | THF | acetone |
|---|---|---|---|---|---|---|---|---|
| PyDPM-PMDA | ++ | ++ | + | + | + | + | + | − |
| PyDPM-BPDA | ++ | ++ | + | + | ++ | ++ | ++ | − |
| PyDPM-ODPA | ++ | ++ | ++ | ++ | ++ | ++ | ++ | − |
| PyDPM-6FDA | ++ | ++ | ++ | ++ | ++ | ++ | ++ | ++ |

NMP, CHCl₃ and THF. It was mainly due to the rigidity of the polymer chain resulting from the relative rigidity of pyromellitic units in the dianhydride moiety. The good solubility of PIs means that films can be prepared at relatively low temperatures without further thermal imidization, and thus can be applied to microelectronics, flexible display, flexible solar cells and other fields.

## 3.4. Wide-angle XRD

The morphological structures of the PIs were observed using wide-angle XRD measurements. As shown in figure 5, the wide-angle XRD patterns of all the films are broad without sharp diffraction signals, which indicates that all PI films have the nature of amorphous state. The amorphous structures of the PIs could be attributed to the introduction of the bulky pyridine ring and ortho-methyl groups in the polymers and resultant loose chain packing. These data also support the high solubility of these PIs.

## 3.5. Thermal properties

The thermal properties of the PIs were evaluated by DSC, DMA, TGA and TMA, and the results are summarized in table 3. The storage modulus and glass transition behaviour of PIs are shown in figure 6. The storage modulus of PIs are in the range of 1.19–1.80 GPa, indicating their high mechanical strength. It could be found that the storage modulus of PyDPM-ODPA slightly increases at 250°C, which may be due to the further imidization. These PIs exhibit high $T_g$s of 359–473°C by DMA and about 20°C lower by DSC. The difference in $T_g$ values by DMA and DSC may be due to different principles of measurement; DMA measures the mechanical response while DSC records the changes in heat flow of polymer. $T_g$ values of these PyDPM-based PIs follow the order: ODPA ≈ 6FDA < BPDA < PMDA. PyDPM-PMDA exhibits the highest $T_g$ of 473°C in DMA test and no glass

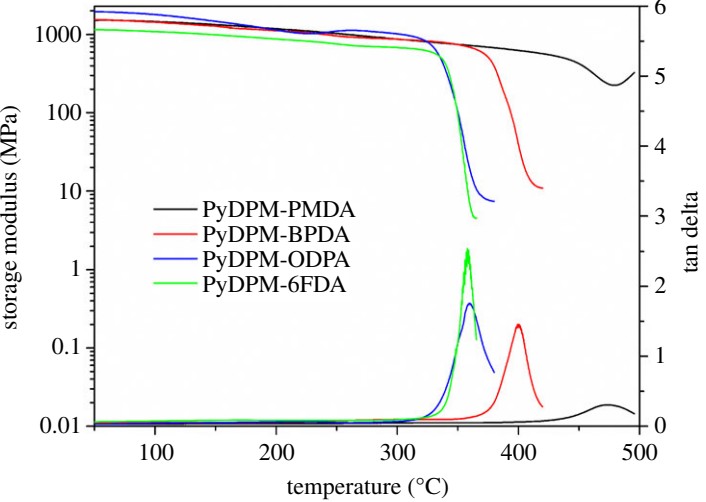

**Figure 6.** DMA traces of PIs.

**Table 3.** Thermal properties of PIs.

| PIs | $T_g^a$ (°C) | $T_g^b$ (°C) | $T_{d5}^c$ (°C) | $T_{d10}^c$ (°C) | char yield[d] (%) | CTE[e] (ppm °C$^{-1}$) |
|---|---|---|---|---|---|---|
| PyDPM-PMDA | —[f] | 473 | 518 | 556 | 69 | 60.8 |
| PyDPM-BPDA | 379 | 400 | 526 | 553 | 66 | 60.3 |
| PyDPM-ODPA | 337 | 359 | 476 | 501 | 62 | 79.2 |
| PyDPM-6FDA | 340 | 359 | 523 | 546 | 63 | 77.6 |
| TPMM-PMDA [39] | —[f] | — | — | 501 | — | — |
| TPMM-BPDA [39] | 378 | — | — | 517 | — | — |
| TPMM-ODPA [39] | 333 | — | — | 519 | — | — |
| TPMM-6FDA [16] | 332 | — | 503 | 520 | — | — |

[a]From the second trace of DSC measurements conducted at a heating rate of 5°C min$^{-1}$.

[b]Measured by DMA with 5°C min$^{-1}$.

[c]Temperature at 5% and 10% weight loss determined by TGA with 20°C min$^{-1}$ under nitrogen.

[d]Residual weight retention at 800°C.

[e]Measured by TMA from 50 to 200°C with 5°C min$^{-1}$.

[f]Not detected.

transition can be found with DSC measurement, which could be ascribed to the rigid pyromellitic residues in the molecular chains. Whereas the lowest $T_g$ value of PyDPM-ODPA (337°C) might be a result of flexible rotary ether bond. Obviously, it could be found that all the PIs exhibited slightly higher $T_g$ values (approx. 5°C) than the analogous PIs derived from TPMM [16,39]. We found that two-step method using our own monomer via ordinary heating cannot produce PIs with high enough molecular weights and the cast films readily break into pieces.

The thermal stability of the PIs was evaluated by TGA measurement under nitrogen. Figure 7 shows the TGA curves of the PIs, and the data are summarized in table 3. These PIs show less than 0.5% weight loss at 250°C, which could be ascribed to a small amount of moisture absorption and residual solvent. The temperature of 5% and 10% weight loss are in the range of 476–526°C and 501–556°C, respectively. Meanwhile, the char yields are over 62% at 800°C. Contrarily to the literatures [33], most of these PIs show higher $T_{d10}$ (approx. 30°C) than analogue PIs from TPMM [16,39]. The thermal stability of PyDPM-based PIs is very close to that of m-PyDPM-based PIs [35], indicating that the introduction of pyridine moiety in the side chain did not deteriorate the thermal stability of PIs.

The dimensional stabilities of PyDPM-based PIs were characterized through TMA and the results are summarized in table 3. Generally speaking, the CTE values of PI are dependent on the extent of the chain

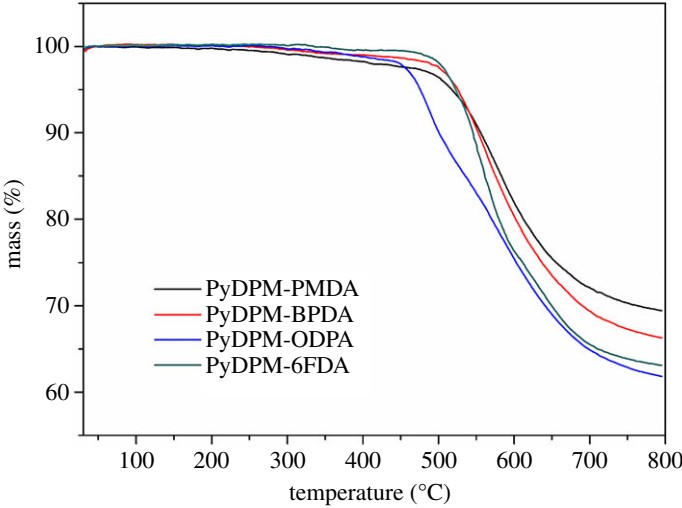

**Figure 7.** TGA curves of PIs.

**Table 4.** Dielectric constants and moisture absorption of PIs.

| PIs | film thickness (μm) | water absorption (%) | dielectric constant | |
|---|---|---|---|---|
| | | | 10 kHz | 10 MHz |
| PyDPM-PMDA | 51 | 3.9 | 4.14 | 3.84 |
| PyDPM-BPDA | 66 | 2.8 | 3.93 | 3.79 |
| PyDPM-ODPA | 56 | 1.7 | 3.93 | 3.81 |
| PyDPM-6FDA | 70 | 1.1 | 3.57 | 3.49 |

packing and linearity of the polymer chains. The higher the linear and stiff backbone structures, the lower the CTE values [40,41]. These PIs show relatively high CTE values of 60.3–79.2 ppm $°C^{-1}$. This is related to the triarylmethane structure, which reduces linearity of the PI chains. What is more, the two ortho-methyl groups destroy the coplanarity of the aryl units, and the pyridine moieties also interrupt the interchain packing. The dianhydrides also have great effect on the CTE values. The rigid linear polymer chains endow PyDPM-PMDA (60.8 ppm $°C^{-1}$) and PyDPM-BPDA (60.3 ppm $°C^{-1}$) the lower CTE value. The higher CTE value of PyDPM-ODPA (79.2 ppm $°C^{-1}$) and PyDPM-6FDA (77.6 ppm $°C^{-1}$) can be explained by the loose molecular chain packing imparted by ether bond or hexafluoroisopropyl.

## 3.6. Dielectric constants and moisture absorption

The dielectric constants and water absorption of the pyridine-based PI are shown in table 4. As pyridine is a highly polar aromatic heterocyclic ring and easily polarized in an electric field, these PIs should have large dielectric constants. However, the two methyl groups in PyDPM increase the free volume and reduce their dielectric constants. PyDPM-PMDA has the highest dielectric constant of 4.14 at the test frequencies of 10 kHz while PyDPM-6FDA has the lowest dielectric constant of 3.57 among them, which may be due to the higher electronegativity of trifluoromethyl groups and larger free volume over pyromellitic units. Furthermore, the water absorption of PI also has a significant influence on the dielectric constant of the films because water has a high dielectric constant ($\epsilon \approx 80$). The highly hydrophobic nature of the fluorine atoms in PyDPM-6FDA also facilitate the lower dielectric constant.

## 3.7. Mechanical properties

All the pyridine-containing PIs can afford flexible, transparent and tough films with thickness of about 60 μm. The mechanical properties are summarized in table 5. All of the PI films show excellent

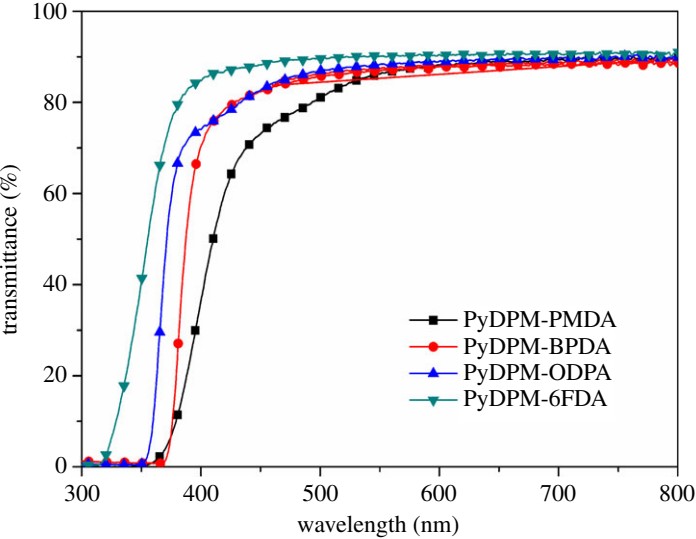

**Figure 8.** UV–Vis spectra of PI films.

**Table 5.** Mechanical properties of PIs.

| PIs | tensile strength (MPa) | tensile modulus (GPa) | elongation at break (%) |
|---|---|---|---|
| PyDPM-PMDA | 93.3 | 3.8 | 12.5 |
| PyDPM-BPDA | 85.8 | 3.4 | 15.2 |
| PyDPM-ODPA | 85.5 | 3.3 | 4.9 |
| PyDPM-6FDA | 82.1 | 3.6 | 3.7 |

mechanical properties. The tensile strength, elongation at break, and Young's modulus of these PI films are in the range of 82.1–93.3 MPa, 3.7–15.2% and 3.3–3.8 GPa, respectively. The mechanical properties of PyDPM-PMDA and PyDPM-BPDA are better than that of others, which might be attributed to rigid structure in the PI backbone.

## 3.8. Optical properties

The optical transparency of PI in the visible wavelength range is a crucial factor in the OLEDs application. The ordinary aromatic PI films always exhibit colours ranging from light yellow to dark brown due to the formation of intra- or intermolecular CTC. The UV–Vis absorption spectra of PI films are shown in figure 8, and the data are summarized in table 6. The cut-off wavelength of PI films is in the range of 313–366 nm, and the transmittance at 450 nm is higher than 82%, except PyDPM-PMDA, indicating that these PI films were almost colourless. PyDPM-BPDA shows light colour and higher $T_g$ because of rigid biphenyl structure. The steric hindrance leads to the destruction of the coplanar structure between biphenyls, so PIs derived from BPDA generally present high $T_g$s and high transparency. PyDPM-ODPA also shows light colour, and it may be due to the introduction of the flexible ether units. Finally, PyDPM-6FDA containing fluorine atoms exhibits the shortest cut-off wavelength of 313 nm and the 88.2% of transmittance at 450 nm. The bulky -CF$_3$ groups of the 6FDA component induced steric hindrance, which interrupted intermolecular chain packing as well as the formation of CTC. Compared with the reference PIs derived from TPMM [39], as shown in scheme 1, these PIs exhibited similar/close optical transparency. This can be explained by the electronegativity nature of pyridine side groups in diamine which can also hinder the formation of intra- and intermolecular CTC, and thus decolour the resulting PIs.

## 4. Conclusion

Four PIs containing pyridine side groups were prepared using a novel diamine monomer PyDPM in this study. It was found that the polymerization via microwave irradiation can afford high-molecular-weight

**Table 6.** Optical properties of PI films.

| PIs | thickness ($\mu$m) | $\lambda_0^a$ (nm) | $T_{400}^b$ (%) | $T_{450}^b$ (%) | $T_{500}^b$ (%) |
|---|---|---|---|---|---|
| PyDPM-PMDA | 23 | 356 | 36.4 | 73.1 | 81.0 |
| PyDPM-BPDA | 21 | 366 | 70.4 | 82.4 | 85.6 |
| PyDPM-ODPA | 20 | 346 | 73.8 | 82.7 | 86.9 |
| PyDPM-6FDA | 20 | 313 | 84.7 | 88.2 | 89.6 |
| TPMM-PMDA [39] | 25 | 350 | 47.1 | — | 85.1 |
| TPMM-BPDA [39] | 22 | 365 | 61.8 | — | 85.7 |
| TPMM-ODPA [39] | 23 | 353 | 70.7 | — | 85.1 |

$^a$UV cut-off wavelength.
$^b T_{400}$, $T_{450}$, $T_{500}$: transmittance at 400, 450, 500 nm, respectively.

PIs. All of the PIs exhibit good solubility in polar and/or non-polar solvents. These PIs present good thermal stability and high heat resistance, with $T_{d5}$ exceeding 467°C in nitrogen and $T_g$s ranging from 358°C to 473°C. PyDPM-BPDA, PyDPM-ODPA and PyDPM-6FDA films exhibit good optical transparency with the cut-off wavelength at 313–366 nm and transmittance higher than 70% at 400 nm. The excellent thermal and optical transmittance can be attributed to one-step synthesis method and the introduction of bulky pyridine rings and ortho-methyl groups. These results indicate these PIs could be potential candidates for optical substrates of OLEDs.

Data accessibility. Data available from the Dryad Digital Repository: https://doi.org/10.5061/dryad.7q704rn [42].
Authors' contributions. K.C., Y.M., Y.Y., S.L. and J.Z. conceived and designed the experiments. K.C., J.H., Y.W. and C.S. performed the experiments. K.C. and Y.M. analysed data as well as wrote the paper. All authors gave final approval for publication.
Competing interests. We declare we have no competing interests.
Funding. This research is supported by the National Natural Science Foundation of China (grants: 51573054 and 21174042) and Student Research Project of South China University of Technology (grant: 9682)

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
