## [Reviewer comments · Royal Society Open Science]

Review History

RSOS-190196.R0 (Original submission)

Review form: Reviewer 1

Is the manuscript scientifically sound in its present form?

Yes

Are the interpretations and conclusions justified by the results?

Yes

Is the language acceptable?

Yes

Is it clear how to access all supporting data?

Yes

Do you have any ethical concerns with this paper?

No

Have you any concerns about statistical analyses in this paper?

No

Recommendation?

Accept with minor revision (please list in comments)

Comments to the Author(s)

The manuscript describes the preparation of new polyimides containing a lateral pyridine moiety. Another interesting point is that they describe the process to obtain the polyimides by a microwave aided reaction in one step reaction process. The manuscript is appropriate and fairly written, the characterization of the as prepared monomer and polymers is properly done. I believe it provides new insight in the possibility of a cleaner and faster process for highly aromatic polyimides preparation. There are only a few minor details that that should be fixed:

In page 5 line 23, they should give the power of the microwave unit used. The same maker provides different powers.

In the thermal properties page 6 line 22 Figure 5. Storage modulus for PyDMP-ODPA present a unstable transition at around 250 C and also at 350. I this an unstable behavior of the sample or is it that there is some solvent left that is leaving the sample at the start of the transition? They should clarify the point.

In Figure 6, page 7 line 15, All TGA thermograms show a weight loss, around 2% at around 250 C. they should explain where this weight loss.

Review form: Reviewer 2

Is the manuscript scientifically sound in its present form?

No

Are the interpretations and conclusions justified by the results?

Yes

Is the language acceptable?

No

Is it clear how to access all supporting data?

Yes

Do you have any ethical concerns with this paper?

No

Have you any concerns about statistical analyses in this paper?

No

Recommendation?

Accept with minor revision (please list in comments)

Comments to the Author(s)

This manuscript reported that optical transparent and soluble polyimide films containing pyridyl side units have been prepared. The chemical structure and various physical and optical

transparency properties of the polymers were clearly characterized. Thus, I think this manuscript can be accepted after revision and double-checking the grammars of the whole manuscript.

1. There are many grammatical mistakes through the whole manuscript. For example, On Line 36 of Page 1, “modified PIs” should be “modify PIs”; on Line 11 of Page 2, the sentence “which may be due to that...” is not correct; on Line 20 of Page 2, “abstacle” is a noun, not verb. Figure 1, “spectra” should be “spectrum”.
2. On page 5, “Figure 1” should be “Figure 4”.
3. Some conclusions were not fully reliable. For example, “Although they reported close data of inherent viscosities, we think our PIs by microwave radiation should have higher molecular weights” there was no direct data to support it.
4. It is not very reasonable that the discussion part about XRD was placed behind the parts involving polymeric thermal, mechanical and other properties.

Decision letter (RSOS-190196.R0)

15-Apr-2019

Dear Dr mo:

Title: Microwave-Assisted Synthesis of High Thermal Stability and Colorless Polyimides Containing Pyridine
 Manuscript ID: RSOS-190196

Thank you for submitting the above manuscript to Royal Society Open Science. On behalf of the Editors and the Royal Society of Chemistry, I am pleased to inform you that your manuscript will be accepted for publication in Royal Society Open Science subject to minor revision in accordance with the referee suggestions. Please find the reviewers' comments at the end of this email.

The reviewers and handling editors have recommended publication, but also suggest some minor revisions to your manuscript. Therefore, I invite you to respond to the comments and revise your manuscript.

Please also include the following statements alongside the other end statements. As we cannot publish your manuscript without these end statements included, if you feel that a given heading is not relevant to your paper, please nevertheless include the heading and explicitly state that it is not relevant to your work. We have included a screenshot example of the end statements for reference.

- Ethics statement

Please clarify whether you received ethical approval from a local ethics committee to carry out your study. If so please include details of this, including the name of the committee that gave consent in a Research Ethics section after your main text. Please also clarify whether you received informed consent for the participants to participate in the study and state this in your Research Ethics section.

OR

Please clarify whether you obtained the necessary licences and approvals from your institutional animal ethics committee before conducting your research. Please provide details of these licences and approvals in an Animal Ethics section after your main text.

OR

Please clarify whether you obtained the appropriate permissions and licences to conduct the fieldwork detailed in your study. Please provide details of these in your methods section.

- Data accessibility

It is a condition of publication that you make available the data and research materials supporting the results in the article. Datasets should be deposited in an appropriate publicly available repository and details of the associated accession number, link or DOI to the datasets must be included in the Data Accessibility section of the article (<http://royalsocietypublishing.org/instructions-authors#question17>). Reference(s) to datasets should also be included in the reference list of the article with DOIs (where available).

Please include a Data Availability section after your main text stating where supporting data are available from, or where they will be made available should your article be accepted for publication.

<http://datadryad.org/submit?journalID=RSOS&manu=RSOS-190196>

- Competing interests

Please include a Competing Interests section after your main text declaring any financial or non-financial competing interests. If you have no competing interests please state 'I/we have no competing interests.'

- Authors' contributions

Please include an Authors' Contributions section at the end of your main text detailing the contribution of each author. All authors should have read and approved the manuscript before submission and this should be stated in the Authors' Contributions section.

The list of Authors should meet all of the following criteria; 1) substantial contributions to conception and design, or acquisition of data, or analysis and interpretation of data; 2) drafting the article or revising it critically for important intellectual content; and 3) final approval of the version to be published.

- Acknowledgements

- Funding statement

Please include a funding section after your main text which lists the source of funding for each author.

Because the schedule for publication is very tight, it is a condition of publication that you submit the revised version of your manuscript before 24-Apr-2019. Please note that the revision deadline

will expire at 00.00am on this date. If you do not think you will be able to meet this date please let me know immediately.

Best wishes,
Dr Laura Smith
Publishing Editor, Journals

On behalf of the Subject Editor Professor Anthony Stace and the Associate Editor Professor John Moses.

Reviewer comments to Author:

Reviewer: 1

Comments to the Author(s)

The manuscript describes the preparation of new polyimides containing a lateral pyridine moiety. Another interesting point is that they describe the process to obtain the polyimides by a microwave aided reaction in one step reaction process. The manuscript is appropriate and fairly written, the characterization of the as prepared monomer and polymers is properly done. I believe it provides new insight in the possibility of a cleaner and faster process for highly aromatic polyimides preparation. There are only a few minor details that that should be fixed:

In page 5 line 23, they should give the power of the microwave unit used. The same maker provides different powers.

In the thermal properties page 6 line 22 Figure 5. Storage modulus for PyDMP-ODPA present a unstable transition at around 250 C and also at 350. I this an unstable behavior of the sample or is it that there is some solvent left that is leaving the sample at the start of the transition? They should clarify the point.

In Figure 6, page 7 line 15, All TGA thermograms show a weight loss, around 2% at around 250 C. they should explain where this weight loss.

Reviewer: 2

Comments to the Author(s)

This manuscript reported that optical transparent and soluble polyimide films containing pyridyl side units have been prepared. The chemical structure and various physical and optical transparency properties of the polymers were clearly characterized. Thus, I think this manuscript can be accepted after revision and double-checking the grammars of the whole manuscript.

1. There are many grammatical mistakes through the whole manuscript. For example, On Line 36 of Page 1, "modified PIs" should be "modify PIs"; on Line 11 of Page 2, the sentence "which may be due to that..." is not correct; on Line 20 of Page 2, "abstacle" is a noun, not verb. Figure 1, "spectra" should be "spectrum".

2. On page 5, "Figure 1" should be "Figure 4".

3. Some conclusions were not fully reliable. For example, "Although they reported close data of inherent viscosities, we think our PIs by microwave radiation should have higher molecular weights" there was no direct data to support it.

4. It is not very reasonable that the discussion part about XRD was placed behind the parts involving polymeric thermal, mechanical and other properties.

Author's Response to Decision Letter for (RSOS-190196.R0)

See Appendix A.

Decision letter (RSOS-190196.R1)

08-May-2019

Dear Dr Mo,

Title: Microwave-Assisted Synthesis of High Thermal Stability and Colorless Polyimides
Containing Pyridine
Manuscript ID: RSOS-190196.R1

It is a pleasure to accept your manuscript in its current form for publication in Royal Society Open Science. The chemistry content of Royal Society Open Science is published in collaboration with the Royal Society of Chemistry.

On behalf of the Subject Editor Professor Anthony Stace and the Associate Editor Professor John Moses.

RSC Associate Editor
Comments to the Author:
(There are no comments.)

Reviewer(s)' Comments to Author:

Appendix A

**To: Editor & Reviewers, Polymer Chemistry -
PY-ART-10-2018-001480**

Dear Prof. John Moses and Dr. Laura Smith,

Please let us sincerely thank you for handling our manuscript and the referees for their precious, constructive comments and suggestions that had helped us to improve our manuscript entitled: “Microwave-Assisted Synthesis of High Thermal Stability and Colorless Polyimides Containing Pyridine” (manuscript RSOS-190196) for *Royal Society Open Science* as a research paper.

Now we have revised the manuscript to address the referees’ concerns in full. Please find below our point-to-point response in which we reiterate the referees’ comments and provide an answer. Major changes in the main text are highlighted in blue ink. Please let us know if you have any further questions or concerns regarding this manuscript.

We sincerely hope that the referees are satisfied with our revised manuscript and it will be acceptable for publication in *Royal Society Open Science*.

Yours sincerely,

Yueqi Mo, Jianqing Zhao

South China University of Technology,

Guangzhou 510640,

P. R. China

List of point-by-point changes and responses to the reviewers' questions

Referee: 1

Comments to the Author(s)

The manuscript describes the preparation of new polyimides containing a lateral pyridine moiety. Another interesting point is that they describe the process to obtain the polyimides by a microwave aided reaction in one step reaction process. The manuscript is appropriate and fairly written, the characterization of the as prepared monomer and polymers is properly done. I believe it provides new insight in the possibility of a cleaner and faster process for highly aromatic polyimides preparation. There are only a few minor details that that should be fixed:

Response: We would like to thank the Referee #1 for her/his positive reviews.

Other comments:

(1) In page 5 line 23, they should give the power of the microwave unit used. The same maker provides different powers.

-- According to the suggestion of the reviewer, we added the microwave power on page 5 as shown below.

of isoquinoline in a glove box. Then, the mixture was transferred to a Biotage Initiator⁺ synthesizer (400W) and irradiated at 80 °C for 2 h, subsequently 230 °C for 30 min, which was automatically adjusted by the machine. After cooling to room temperature,

(2) In the thermal properties page 6 line 22 Figure 5. Storage modulus for PyDMP-ODPA present a unstable transition at around 250 C and also at 350. I this an unstable behavior of the sample or is it that there is some solvent left that is leaving the sample at the start of the transition? They should clarify the point.

-- Thanks the referee very much. According to the suggestion of the reviewer, we

retested this sample and found that the transition at around 250 °C disappeared this time. So we think the transition at around 250 °C was originated from the unstable instrument at that time. The new DMA traces were showed below. The storage modulus of PyDPM-ODPA was smoothly decreases before 220 °C, and then slightly increases at 250 °C, which could be ascribed to the further imidization.

PIs were shown in Figure 6 Fig-5 displays the DMA trace of the PIs. The storage modulus of PIs are in the range of 1.19-1.80 GPa, indicating their high mechanical strength. It could be found the storage modulus of PyDPM-ODPA slightly increases at 250 °C, which may be due to the further imidization. These PIs exhibit

(3) In Figure 6, page 7 line 15, All TGA thermograms show a weight loss, around 2% at around 250 °C. they should explain where this weight loss.

-- According to the suggestion of the reviewer, we checked the TGA data and found all the PIs showed less than 0.5% weight loss at 250 °C. This may be related to the membrane preparation process. In order to get a colorless or light color film, we only thermally baked in an oven at 80 °C for 12 h and 200 °C for 12 h to remove the solvent. So we think the weight loss may be ascribed to a small amount of moisture absorption, especially the residual solvent.

Referee: 2

Comments to the Author

This manuscript reported that optical transparent and soluble polyimide films containing pyridyl side units have been prepared. The chemical structure and various physical and optical transparency properties of the polymers were clearly

characterized. Thus, I think this manuscript can be accepted after revision and double-checking the grammars of the whole manuscript.

Response: We would like to thank the Referee #2 for her/his positive reviews.

Other comments:

(1) There are many grammatical mistakes through the whole manuscript. For example, On Line 36 of Page 1, “modified PIs” should be “modify PIs”; on Line 11 of Page 2, the sentence “which may be due to that...” is not correct; on Line 20 of Page 2, “abstacle” is a noun, not verb. Figure 1, “spectra” should be “spectrum”.

-- According to the suggestion of the reviewer, we have corrected over 20 grammatical mistakes of the manuscript. Please check the highlighted manuscript.

(2) On page 5, “Figure 1” should be “Figure 4”.

-- Actually, we checked ^1H NMR spectra of PyDPM and PyDPM-BPDA and found nothing wrong with Figure 1 and Figure 4.

(3) Some conclusions were not fully reliable. For example, “Although they reported close data of inherent viscosities, we think our PIs by microwave radiation should have higher molecular weights” there was no direct data to support it.

-- According to the suggestion of the reviewer, we deleted this sentence. Furthermore, we also checked the other related conclusions to assure that they are reliable enough.

(4) It is not very reasonable that the discussion part about XRD was placed behind the parts involving polymeric thermal, mechanical and other properties.

-- According to the suggestion of the reviewer, we placed the XRD discussion ahead of thermal properties.